# Soft Contact Lenses as Drug Delivery Systems: A Review

**DOI:** 10.3390/molecules26185577

**Published:** 2021-09-14

**Authors:** Iwona Rykowska, Iwona Nowak, Rafał Nowak

**Affiliations:** 1Faculty of Chemistry, Adam Mickiewicz University, Uniwersytetu Poznańskiego 8, 61-614 Poznań, Poland; grzesiw@amu.edu.pl; 2Eye Department, J. Strus City Hospital, Szwajcarska 3, 61-285 Poznań, Poland; raf.nowak@wp.pl

**Keywords:** contact lenses, drug delivery, drug-controlled release, drug delivery systems based on contact lenses in ophthalmic therapies

## Abstract

This review describes the role of contact lenses as an innovative drug delivery system in treating eye diseases. Current ophthalmic drug delivery systems are inadequate, particularly eye drops, which allow about 95% of the active substance to be lost through tear drainage. According to the literature, many interdisciplinary studies have been carried out on the ability of contact lenses to increase the penetration of topical therapeutic agents. Contact lenses limit drug loss by releasing the medicine into two layers of tears on either side of the contact lens, eventually extending the time of contact with the ocular surface. Thanks to weighted soft contact lenses, a continuous release of the drug over an extended period is possible. This article reviewed the various techniques to deliver medications through contact lenses, examining their advantages and disadvantages. In addition, the potential of drug delivery systems based on contact lenses has been extensively studied.

## 1. Introduction

Ophthalmic drug delivery has always been a challenge for ophthalmologists and scientists from a variety of disciplines. It is estimated that the bioavailability of ophthalmic drugs is uncertain and is about 5% or less. This is a consequence of anatomical and physiological barriers, including tear drainage and epithelial transport limitations. Unique static and dynamic eye barriers exclude the penetration of xenobiotics and discourage the active absorption of therapeutic agents. Designing an ideal delivery regimen should involve increased bioavailability and controlled drug release at the target tissue, overcoming the ocular barriers [1,2].

Eye medications administered in the conventional form of eye drops or ointments are often characterised by low bioavailability. In addition, they require repeated daily administration, which, combined with low patient compliance, causes doses to be avoided or administered incorrectly, contrary to therapeutic recommendations. Attempts to increase the bioavailability of ophthalmic medicines by using various modern solutions such as viscous solutions, suspensions, emulsions, ointments, gels, polymer inserts, and colloidal systems are still unsatisfactorily challenging in pharmaceutical research [3]. Hence, the use of contact lenses as drug delivery systems has been increasingly explored in recent years [4].

The main objectives for the development of DCR (drug-controlled release) based on SCLs (soft contact lenses) are [5]:to increase the drug delivery efficiency;to improve patient compliance and reduce undesirable systemic side effects, especially in chronic diseases such as glaucoma and dry eye;to enhance SCLs tolerance, particularly in patients affected by dry eye syndrome and ocular allergies;to design “bandage contact lenses” modified with antimicrobial or anti-inflammatory agents for managing corneal wound healing.

Offering greater drug bioavailability to the cornea than conventional topical formulations, SCLs seem to be the ideal ophthalmic drug reservoir [6].

The idea of using contact lenses as a carrier of active ingredients is a relatively new strategy that is still being developed and improved. The dynamic of this research is evidenced by the number of publications that appear in scientific journals. Table 1 summarises the review papers that have been published in recent years [1,2,3,4,5,6,7,8,9,10,11,12,13,14,15,16,17,18,19,20,21,22,23,24,25,26,27,28,29,30,31].

According to the authors (Table 1), soft contact lenses are highly desirable as an effective eye drug delivery system. Authors also stress that further research is still needed to achieve an effective, safe, and stable product to introduce to the pharmaceuticals market [4].

Contact lenses are separated from the cornea by a thin layer of fluid called the post-lens tear film. This fluid is poorly mixed with the rest of the ocular tear film. The mixing time with the external tear fluid covering the outer surface of the contact lens is estimated at approximately 30 min [32]. For this reason, ophthalmic drugs released from SCL may remain on the corneal surface for at least 30 min. This is about 15 times longer compared to regular eye drops [14]. Prolonged contact time of the drug with the cornea can increase its bioavailability up to 50% compared to the 1–5% efficiency of eye drops [33]. Moreover, improved bioavailability implies a lower amount of drug in the systemic circulation, which reduces the risk of potential side effects. The attractiveness of DDSCL (drug delivery soft contact lenses) as reservoirs of ophthalmic drugs results in numerous studies developing systems with a high drug loading capacity and balanced and controlled medicine release [10].

## 2. Contact Lenses as Drug Delivery Systems

Contact lenses are hard or soft polymer devices designed to fit the cornea to correct refractive errors. They can be made of hydrophilic or hydrophobic polymers. Hydrogel contact lenses appear to effectively deliver drugs to the eye since they better absorb aqueous solutions [12]. Contact lenses offer higher bioavailability of the drug than other non-invasive ophthalmic medications, such as drops or ointments, due to the proximity of the contact lens to the cornea. They also provide a significant advantage in dosage over topical eye drops [1].

There are two main groups of contact lenses depending on the designed material: soft contact lenses–made of hydrogel or silicone hydrogel polymers, and rigid gas-permeable contact lenses (RGP) [28]. The soft materials for use as drug delivery systems are of more interest because of their hydrophilic properties, biocompatibility, and comfort of use [34,35]. For this reason, SCLs account for 87% of matches in clinical practice, as opposed to 13% of RGP contact lenses [36,37].

The physical and chemical properties of the polymers used are essential in the design and quality control of DDSCL. Polymers’ most critical physical properties for drug-releasing contact lenses are transparency, oxygen permeability, glass transition temperature, wettability, and water content [9].

## 3. Soft Contact Lenses Parameters

### 3.1. Transparency

The transparency of the contact lens is a crucial parameter determining its functionality and must not be impaired by the added drug. DDSCL developed using novel techniques such as molecular or supercritical solvent imprinting, liposome loading, and microemulsions showed good transparency [38].

### 3.2. Oxygen Permeability

Low oxygen transfer through the contact lens can result in serious side effects. Since the human eye is insufficiently oxygenated by the system of blood vessels, and the oxygen supply is mainly carried out through exposure to air, oxygen delivery and effective carbon dioxide removal must be carried out through the contact lens, ensuring gas circulation. Low-oxygen-transmissible SCLs further impede oxygen flow to the cornea with possible loss of corneal transparency upon overwear. The SCL oxygen permeability is defined as Dk (D multiplied by k), where D is the diffusion coefficient, and k is the partition coefficient of oxygen in the lens material.

The gas permeability of soft contact lenses has been improved by silicone-based polymer hydrogel lenses made of polydimethylsiloxane (PDMS). PDMS exhibits impressive permeability (Dk = 600 barrers) while maintaining comfort, wettability, and biofilm resistance compared to silicone-based hydrogel lenses. However, the long-worn contact lens must ensure oxygen permeability of not less than Dk > 87 barrers to avoid corneal hypoxia. Achieving Dk with conventional hydrophilic contact lenses on such a level is very difficult [9].

### 3.3. Glass-Transition Temperature

The glass transition temperature (Tg) is the temperature below which the physical properties of polymers change to those of a glassy or crystalline state. The Tg of contact lens material is expected not to change due to modification by the drug or application of other additives. Numerous studies show that different production methods of drug-delivering contact lens materials do not affect the Tg values. For example, the change in Tg after the addition of β-CD (β-cyclodextrin) was insignificant, suggesting that the addition of β-CD has little or no effect on the degree of hydrogel cross-linking or network stiffness [13]. Costa et al. [39] also found that the Tg of SCLs does not change due to the impregnation and release of the drug. On the other hand, Yanez et al. found no changes in the Tg values of SCL in the subsequent supercritical stages of fluid processing [40]. Therefore, it can be concluded that most of the DDSCL assembling methods allow the design of medicine charged lenses without changing their material’s Tg [39].

## 4. Wettability

Contact lens wettability is a critical variable that affects its physiological compatibility and the tear fluid’s stability on the lens’s surface. The wetting capacity of soft contact lenses is parameterised by the contact angle value. The contact angle is the angle between a drop of liquid and the surface of the contact lens. Values of contact angles above 90° mean that the lens surface has poor wettability. The texture is entirely nonwetting when the angle exceeds 180°.

Hydrophobic polymers repel water molecules, which are the main component of the tear fluid. Such materials disrupt the flow of tears and cause the albumin membrane to settle on the lens, which can eventually cause eye infection and/or irritation. For this reason, the contact lens should be highly hydrophilic and resistant to biofilm deposition during long-term use. To date, there are several DDSCL preparation techniques known that do not significantly impair lens wettability. Thus, poly-2-hydroxyl ethyl methacrylate (pHEMA) wettability has been slightly increased due to glycosyl methylate (GMA) copolymerisation instead of a slight reduction when β-CD was added [39,41]. On the other hand, no significant change in the contact angle value was recorded when timolol maleate and acetazolamide were applied by the supercritical solvent impregnation technique [3].

### The Water Contents

The water content of contact lenses is an essential factor in increasing the comfort of their use. Higher water content means better gas exchange of pHEMA lenses [10]. The gas permeability of a contact lens is proportional to the amount of water in the lens. As the content of water in the lens increases, the permeability increases linearly. Thanks to this, soft contact lenses are characterised by excellent oxygen permeability that may allow long-term use without the risk of eye dysfunction.

On the other hand, increasing the water content of a contact lens can decrease binding interactions and weaken the polymer network. This, in turn, can increase the vulnerability of the lens surface to scratches. Consequently, a softer lens provides less protection for the cornea.

Kim et al. [42] showed no correlation between the amount of drug released and the water content of the lens for a hydrophobic drug. Lenses with a higher water content absorb more water-soluble drug compounds and release them later into the tear film. The likely reason is that hydrophobic drugs will split into silicone-rich phases, and the drug partition coefficient in gels is mainly affected by the silicone composition [43].

It was shown that the incorporation of functionalised compounds into hydrogels could change oxygen permeability, water content, wettability, stiffness, flexibility, glass transition temperature, and light transmittance [44]. These are critical parameters affecting the comfort and quality of vision and safety of contact lenses [45]. El Shaer et al. [46] found that the inclusion of prednisolone nanocapsules worsened the parameters above.

## 5. Manufacturing Materials

The commercialisation of DDSCL requires contact lenses with legally compliant physical and chemical properties that provide comfort of use and good vision quality [28,47,48].

The most-used monomers to produce SCLs are 2-hydroxyethyl methacrylate (pHEMA) and N-vinylpyrrolidone (NVP). These monomers show suitable properties for safe use on the ocular surface, such as high oxygen permeability, optimal water content, and appropriate wettability [34,35]. Their mechanical properties allow soft contact lenses to fit the shape of the eyeball [42].

## 6. Drug Carriers

Typical polymeric carriers for therapeutic contact lenses include drug-containing polymer nanoparticles and polymeric implants inside contact lenses. Table 2 summarises the characteristics of various polymers as therapeutic carriers for contact lenses [19].

## 7. The Role of Hydrogels in DDSCL

Hydrogels are polymer networks having hydrophilic properties. They have been proposed as drug delivery systems to overcome tissue barriers and ocular protection systems, such as blinking and tear drainage. They have been developed as stand-alone systems or supported by other technologies, e.g., nanotechnologies.

The hydrogel drug delivery systems can be formulated from both synthetic and natural polymers known as biopolymers. Biopolymers are at the forefront of many research efforts in delivering drugs to the eye. They are biodegradable, biocompatible, and non-cytotoxic, which make them very useful for ophthalmic treatment.

The favourable biodegradation profiles of biopolymers allow their use without the risk of inducing inflammation. There are also many studies illustrating their low toxicity profiles [25]. Biopolymers are readily available and relatively inexpensive compared to some synthetic polymers.

Biopolymer-based hydrogels as drug delivery systems associated with nanoparticles, nanoliposomes, and nanowires provide an opportunity to develop solutions for an adequate and sustainable supply of ophthalmic drugs. There is still a need to establish topical drug delivery systems, including hydrogel-based ones [26].

Early approaches to contact lens-assisted drug delivery relied on the absorbance of a drug solution during pre-wear soaking. For the first time, soft contact lenses as drug delivery systems were proposed in 1965 by Sedlácek [49]. For this purpose, contact lenses were soaked with a 1% homatropine solution. The pupil dilation was significantly more effective in patients treated with DDSCL. Over the next decade, various studies on dip-coating techniques were conducted, including the use of pilocarpine to treat acute glaucoma [50,51,52,53]. McNamara et al. [54] found that tear exchange while wearing modified soft contact lenses took about 30 min, compared with 5 min when the medicine is dropped directly to the eye [28].

In DDSCL, the principle is that if the aqueous contact lens wetting solution contains enough pharmaceutically active material, the drug diffuses from the polymer matrix into the ocular tear film and interacts with the eye tissue. However, it turns out that contact lenses prepared by this method have a limited potential for drug delivery. The drug supply is carried out in an uncontrolled manner, giving the initial “burst release” of the drug, then a rapid decrease of the medicine in a relatively short time is observed [12,55,56].

To ensure patients comfort, different types of lenses are available on the market. Thus, it is possible to adjust SCLs to the preferences and needs of the user, as well as to adapt to the patient’s lifestyle. Given the mode of wearing, contact lenses can be divided into daytime and extended mode lenses or lenses for continuous wear [54,57]. For this reason, the dynamics of the drug release must be tuned not only to the therapeutic requirements but also to the period of lens wearing. As it turns out, there is still a shortage of therapeutic devices based on SCLs with the desired drug release profile, which prompts researchers to further inquiries on this topic.

Figure 1 shows the distribution of the drug in the eye depending on the method of application.

Since the molecular interaction between commercially available hydrogels and medicines is not specific, the retention time of drug-soaked contact lenses is limited by the physical properties of the lenses, mainly iononicity and water content [43]. In addition, to reduce the risk of ocular surface irritation and optimise the diffusion of the drug through the cornea, a pH ranging from 6.6 to 9.0 [58] should be maintained.

## 8. Drug Loading Techniques

In recent decades, attempts have been made to extend drug residence time and improve the bioavailability of various lens-based drug delivery systems. SCLs surface modifying methods include dip-coating (soaking); diffusion barrier insertion (Vitamin E); incorporation of functional monomers, ligands, and a polymeric matrix; molecular imprinting, cyclodextrin vaccination [59], incorporation of colloidal, drug-loaded nanoparticles or other colloidal nanostructured systems; and surface coating by multilayer film deposition of colloidal nanoparticles or ligands [5,12,28,58,60] (Figure 2).

The following new drug delivery systems based on SCLs that improve bioavailability, solubility, penetration, and retention of ophthalmic drugs will be discussed [5,12].

### 8.1. Soaking (Dip-Coating)

Soaking soft contact lenses in a drug solution is the simplest method of applying drugs to the hydrogel matrix (Figure 2, first step). The main advantage of this method is the possibility of modifying commercially available pHEMA contact lenses [35]. Its limitation, however, is that the release of the drug occurs very quickly since the charging of the drug is dependent on the ionity and water content of contact lens materials [61,62]. This method, developed in 1965 [49], has been used in many studies to incorporate ophthalmic drugs such as antibiotics [63], antihistamines [64], non-steroidal anti-inflammatory drugs, corticosteroids [65], tear stimulants [61,62], and anti-glaucoma medications [66]. Even though most of the studies in the literature have been performed in vitro, some authors have assessed both the retention time and efficacy of various drugs in vivo, showing better results for soaked contact lenses than topical eye instillation.

It was also found that the early inclusion of Vitamin E in the hydrogel matrix creates a diffusion barrier that prolongs the release time of the drug. Hsu et al. [67] increased the retention time of thymol in the pHEMA-based contact lens from 1 h (without Vitamin E) to 25 h (with 20% Vitamin E) [28].

In addition, Vitamin E, as a powerful antioxidant, protects the cornea from UV radiation and prevents the oxidation of many sensitive drugs [67,68,69,70]. However, the use of Vitamin E has certain limitations, including a reduction in ion and oxygen permeability, a change in mechanical CL properties and protein adsorption on the CL surface due to the hydrophobic nature of Vitamin E [15].

### 8.2. Functional Monomers Incorporation

The incorporation of functional monomers changes the physical properties of the hydrogels. Modifying CLs by introducing functional monomers (Figure 2, second step) can be achieved with two different strategies.

The first is the addition of ionic compounds in the polymerisation process. Thus, strong attachment points are formed between the contact lens and the drug. Most often, the ionity of hydrogels is modified by adding acryl-vinyl derivatives to the pHEMA [71]. Cationic monomers, such as methacrylic amino propyl-trimethyl ammonium chloride (MAPTAC), increase the load of anionic drugs and delay their release [72]. Others, such as 2-methoxy ethyl phosphate (MOEP) or methyl methacrylate (MMA), are used to build up cationic drugs [73,74,75]. DDCLs loaded with various ophthalmic drugs were prepared using this method, and in vitro studies were conducted. This extended the drug retention time from several hours to two months [15,76,77,78,79,80].

In the case of hydrophobic drugs, the second approach of functionalised hydrogels with cyclodextrins is used.

Cyclodextrins are a family of cyclic oligosaccharide with a hydrophobic cavity capable of carrying small molecules of drugs [81]. The functionalised cyclodextrin can be linked to pHEMA by acrylic/vinyl assisted copolymerisation or glycidyl methacrylate (GMA) cross-linking. Some in vitro studies have shown that drug release from cyclodextrins can last from several hours to two weeks [35,76,82,83]. Li et al. [84] showed that the release of diclofenac sodium was higher in rabbits compared to in vitro conditions for the first hour (73% and 42%, respectively) and lasted longer (three days and two days, respectively) [28].

### 8.3. Molecular Imprinting

Molecular imprinting creates a strong interaction between contact lenses and drugs using a drug template formed during polymerisation. This way, imprints of cavities of the appropriate size are created in a polymer, which can be further filled up with drugs (Figure 2, third step). This method uses functional monomers, mainly acrylic-vinyl derivatives, to correspond to drugs’ molecular conformation [85]. The addition of highly homogeneous monomers can make drugs easily bind to contact lenses, but there is a risk that release may be uncontrolled. For this reason, it is imperative to control the affinity ratio of functional monomers to drugs to obtain consistent and controlled release over time [86]. Molecular imprint technology is a technique for modifying a polymer, increasing its affinity for drug molecules, thereby enhancing the loading potential of drugs and extending their delivery time [87,88,89].

In 2002, Hiratani imprinted soft contact lenses using methacrylic acid (MAA) as functional monomers to incorporate timolol into N, N-diethyl acrylamide (DEA) and pHEMA hydrogels [56,90,91]. Alvarez-Lorenzo et al. [92] used the same strategy to produce pHEMA contact lenses with the addition of norfloxacin. They reported that the reservoir capacity increased 300 times compared to pHEMA lenses without molecular imprinting technology [12]. Both studies showed in vitro release of more than 12 [93] and 24 h [90]. In turn, the Alvarez-Lorenzo study with SCLs molecular imprint for timolol showed a 70% increase in absorption [90]. Following these results, Hiratani et al. tested SCLs in rabbits. The authors showed that the drug release time was doubled compared to conventional lenses (180 vs. 90 min) [94].

The presented method has been used to load other ophthalmic drugs such as antibiotics [95] and antimicrobial [96], antihistamines [97], antiallergic [82,98], non-steroidal anti-inflammatory drugs [99], corticosteroids [100], therapeutic agents that can increase eye comfort [101], also used for diabetes patients [100,102], and humectants in the treatment of dry eye [103] and, according to the latest research, with antiviral drugs [104].

The article [97] by Tieppo et al. reported the successful results of in vivo sustained release of the low molecular weight antiallergic therapeutic agent ketotifen fumarate from molecular imprinted therapeutic CLs. Soft poly (pHEMA-co-AA-co-AM-co-NVP-co-PEG200DMA) contact lenses were prepared. A sustained, effective drug concentration was maintained in the contact lens tear film for an extended time.

### 8.4. Colloidal Nanoparticles Incorporation

The development of nanomaterials allowed the encapsulation of drugs inside colloidal nanoparticles such as liposomes, micelles, microemulsions, and polymer nanoparticles (Figure 3).

The incorporation of drug-containing nanoparticles into the polymer matrix of the contact lens is an effective strategy for sustained drug delivery. This approach allows sustained drug release to be adjusted to the patient’s needs, from a few hours to several weeks. Nanoparticles (10 to 100 nm) act as a barrier against metabolic degradation during the release of drugs to the eye surface [105]. Through chemical bonds, functionalised nanoparticles can be incorporated in soft contact lenses by the polymerisation reaction, infiltration, or immobilisation on the contact lens surface [35]. It has been shown that nanoparticle-functional contact lenses demonstrate significantly longer retention times than eye drops containing drug nanoparticles [35,105].

Over the past decade, the following nanoparticle medicines have been applied to soft contact lenses: antibiotics [106], antihistamines [107], immunosuppressants [108], corticosteroids [46], and glaucoma [109].

In animal studies, an increase in the release time of the drug up to seven days for chitosan nanoparticles with cyconazole [110] was observed; 10 days when ketotifen loaded onto silica nanoparticles was used in rabbits [107]; and 14 days when cyclosporine A loaded onto polymer nanoparticles was tested on mice [108].

Therefore, lenses with a layer of fibrin or PLGA provide admittedly long-term drug delivery benefits, but this technique makes lenses opaque. A clear “window” in the centre of the lens must be preserved to ensure good vision during treatment [92,111,112,113].

Moreover, Gulsen and Chauhan [114] conducted a pilot study to determine the effectiveness of pHEMA modified with nanoparticles. The nanoparticles were based on an oil-in-water microemulsion filled with lidocaine, a hydrophobic drug; the droplets were encapsulated in a silica coating stabiliser. The nanoparticles were incorporated into the hydrogel matrix during polymerisation. Hydrophobic lidocaine has low and finite water solubility; therefore, it can slowly diffuse from the nanoparticles into the aqueous phase of the gel matrix and further into the tear film. The nanoparticle-loaded hydrogels remained clear. In vitro studies showed an initial burst followed by a slow and steady release. By day ten, virtually all the drug was released. The authors concluded that nanoparticle-loaded hydrogels might be suitable for controlled drug delivery over several days at therapeutically effective concentrations.

Gulsen and Chauhan [115] continued the nanoparticle modified pHEMA research developing four micro emulsion-based formulations. pHEMA types 1 and 2 were based on rapeseed oil with Tween^®^ 80 and Panadon SDK, with or without a stabilising silica coating. Types 3 and 4 were based on Brij^®^ 97 hexadecane with or without a stabilising silica coating and lidocaine as a model drug. Formulation type 1 was opaque due to the poor solubility of Tween^®^ 80 in pHEMA. Formulation type 2 lost some transparency, indicating that the silica coating reduces the interaction between the surfactant and the pHEMA. Type 3 showed a minimal reduction in clarity but was not as transparent as clean pHEMA. Type 4 showed no appreciable loss of transparency due to silica coating stabilisation. In vitro release studies showed that the hexadecane-based formulations of Brij^®^ 97 were suitable for sustained drug delivery at a therapeutic rate of up to 8 days. The formulation based on Tween^®^ 80 was considered unsuitable due to poor particle stability and aggregation. Gulsen and Chauhan speculate that this work will develop particle-based systems that could respond to changes in pH or temperature, reduce burst releases, and optimise release rates [114,115].

Due to lens opacity problems when using surfactants with pHEMA, Jung and Chauhan [116] proposed a contact lens system based on nanoparticles with timolol and pHEMA, made without surfactants. Using thermal polymerisation techniques, they formed drug-containing nanoparticles based on cross-linking monomers, propoxylated glycerol triacrylate (PGT), and ethylene glycol methacrylate (EGDMA) and incorporated them into pHEMA hydrogels. Their product was a transparent drug-impregnated hydrogel with a temperature-dependent release rate of two to four weeks. The drug kinetics appeared to be temperature-dependent. Therefore, it could not be removed during storage and was activated when placed on the cornea [12].

The same authors conducted further studies with lenses with PGT-timolol, verifying the release up to one month without interfering with optical properties, oxygen, or ion permeability [117]. Similarly, they conducted a preliminary in vivo study that showed a significant reduction in intraocular pressure for two days.

### 8.5. Drug-Polymer Film Embedded

Drug-polymer film embedded contact lenses are another innovation aimed at increasing the retention time of the drug (Figure 2). Coating polymers used to bind drugs to pHEMA are polylactic glycolic acid (PLGA) [62], polyvinyl alcohol (PVA) in combination with chitosan [118], or ethyl cellulose (CE) in combination with Eudragit S-100^®^ [119]. An essential aspect of this method is not to disturb the transparency of modified contact lenses. For drugs that do not have hydrogel-like properties, it is possible to synthesise the medicine film on the periphery of the lens, thus not affecting the optical zone.

It was shown that the drug release effectiveness is directly proportional to the thickness of the drug film [118,119,120]. In vivo, sustained drug release was achieved when nonsteroid anti-inflammatory drugs were applied (12 h), two days for antibiotics and antihistamines, or seven days for corticosteroids [28,120,121,122,123].

Hyatt et al. [111] investigated the release profiles of gentamicin and vancomycin from fibrin-coated contact lenses. The goal was to develop a system that could offer controlled and sustained drug delivery for at least 8 h. They revealed that the fibrin lens systems performed better for the sustained delivery of gentamicin than the impregnated plain lenses. However, effectiveness in delivering vancomycin worsened compared to the basic impregnated lenses. Fibrin-containing lenses have demonstrated the potential to treat bacterial keratitis.

Introducing a drug-polymer film of PLGA (polylactic glycolic acid) into the SCLs matrix was made by Ciolino et al. by loading latanoprost between two layers of silicone hydrogel [120]. In vivo studies in rabbits showed therapeutic latanoprost levels for four weeks, increasing the intravitreal concentrations 30-times compared to treatment with drops. The same author conducted a study in monkeys with glaucoma, obtaining sustained therapeutic values for over eight days. Moreover, the therapeutic effect was more significant than administering eye drops containing the drug [124].

Ciolino et al. [112,113] also found that contact lenses with PLGA film retained antifungal properties for up to three weeks in vitro. The prototype ciprofloxacin-releasing contact lens showed controlled release at therapeutically active concentrations for up to four weeks in vitro [12].

Xu et al. [109] also studied SCLs loaded with micelles using MPEG-PLA (methoxy poly (ethylene glycol)-poly (lactide) copolymer) loaded with timolol and latanoprost. Intraocular pressure decreased within seven days when administered to rabbits with ocular hypertension, showing significantly better results than eye drops [109]. However, the use of oil disturbs the lens’s optical properties. The solution is to make a film-free area in the optical zone [28,124].

### 8.6. Supercritical Fluid

Supercritical fluids are all compounds that reach pressure and temperature conditions above their critical point. Hydrophilic and hydrophobic drugs can be readily dissolved in supercritical solvents for use in a soft contact lens matrix (Figure 2). Drug loading is achieved by soaking the lenses in a supercritical solvent-drug solution under controlled conditions [39] or assisted by molecular imprinting [15]. The advantage of the first method is that commercial soft contact lenses can be used, while the protocol containing molecular imprinting requires an initial polymerisation reaction. It has been shown that both ways are more effective than conventional soaking in terms of the amount of drug released [39,125].

However, other literature studies have shown that supercritical solvents give fewer promising results. So far, it has been possible to extend the drug retention time to only a few hours [15,39,126]. This finding should be carefully considered as further research is needed.

### 8.7. Guest-Host Complexes

Cyclodextrins (CDs) are a promising drug delivery system due to forming an inclusion complex with various drug molecules in solution and the solid-state [127]. Using “container molecules” as a drug reservoir can increase the ability to deliver drugs by hydrogel contact lenses. For this purpose, cyclodextrins with “guest-host” properties were tested.

The complexation of cyclodextrins and drug molecules is a dynamic process, accompanied by weak covalent bonds. As a result, increased uptake, and bioavailability enhanced tolerance and decreased cytotoxicity of the ophthalmic drug are observed [124,128,129,130,131].

The chemical combination of dextrins with contact lens hydrogels can offer many alternatives for a drug delivery device. This combination was first used as a carrier for acetazolamide, showing sustained drug release for up to 24 days. Moreover, this release can be controlled by changing beta-CD concentrations during copolymerisation [128].

Similarly, other types of dextrins (alpha, beta, and gamma dextrins) have been tried to improve DCRS. Thus, polydextrins have been shown to facilitate the loading of high concentrations of ethoxzolamide (another carbonic anhydrase inhibitor) while providing sustained release over several weeks [132].

These results are encouraging, especially considering that many drugs have been administered mainly orally so far, limiting their therapeutic effect on the eye. The proposed solution may allow for a drastic reduction in the therapeutic dose of the medicine and thus a decrease in side effects [28].

The strategy used by dos Santos et al. [128] was to synthesise a methacrylated β-cyclodextrin to produce a copolymer with pHEMA and EGDMA. The polymers formed were transparent. Drug loading was achieved by soaking anhydrous polymers in acetazolamide or hydrocortisone solutions for four days. The performance of hydrogels was tested in vitro. The authors observed a controlled drug charging rate and several-day drug release. They continued their study by developing a hydrogel formulation using β-cyclodextrin implanted on pHEMA and coglycidyl methacrylate (GMA). This system increased the diclofenac loading by 1300% and continued releasing the drug (two weeks) into the tear fluid [133].

Xu et al. [134] produced hydrogel films implanted on pHEMA contact lenses made of mono methacrylated β-cyclodextrin and trimethylolpropane trimethacrylate. Puerarin was incorporated as a model drug by soaking the prepared hydrogel in the drug solution. In vivo studies in rabbits showed that the hydrogels modified by guest-host complexes provided sustained drug release with better efficacy than commercially available puerarin eye drops. Such SCLs also had excellent mechanical properties. On this basis, it was considered that the proposed material is suitable for the supply of drugs with reusable contact lenses [12].

## 9. DDSCL in Ophthalmic Treatment

Eye diseases are a common problem that affect a significant portion of the human population. Therefore, there are many types of ophthalmic drugs on the market. However, their use can be troublesome, as mentioned many times before. Extensive research on a formulation that extends the residence time and the bioavailability of drugs on the ocular surface is still ongoing.

SCLs as DCRS, appear as an effective treatment option for a variety of eye dysfunctions. They provide the timely and effective release of drugs in corneal epithelial defects, help treat infections, and promote local healing [5]. The goal of such systems is to improve the bioavailability and efficient local delivery of medicines. In this way, patients’ compliance is improved, and the doses of drugs are reduced. The possibility of covering the lenses with wetting agents also ensures better safety and convenience of their use.

### 9.1. Glaucoma

The use of contact lenses as a reservoir of drugs is a promising treatment system for chronic eye diseases [135,136,137,138], which is vital for glaucoma patients. Glaucoma is a chronic disease that mainly affects the elderly, who often have reduced manual ability to administer eye drops. Such treatment needs appropriate regularity. These factors increase the likelihood of noncompliance [139,140], leading to therapeutic failure [141,142,143]. The literature shows that the most popular preparation techniques for glaucoma DCRS are simple soaking, drug-loaded colloidal nanoparticles, and molecular imprinting.

The first DCR glaucoma systems concerned pilocarpine applied to SCLs by the soaking-and-release method [50]. Subsequently, other modifications were made to obtain better SCLs based devices [52,54,144,145,146,147]. Ultimately, the same treatment effects were achieved after two hours of wearing contact lenses with 1% pilocarpine and 4% pilocarpine installation [51]. The same strategy was also used with timolol and brimonidine, showing equivalent IOP (intraocular pressure) control using SCLs only 30 min per day for 14 days [148]. Moreover, the use of SCLs reduces the likelihood of undesirable systemic side effects [149].

This approach has also been applied to novel systemic glaucoma treatments, such as melatonin and its analogues application to load them on SCLs. A balanced release of the drug in the first 30 min was achieved. For dinucleotides such as Ap4A and Gp4G, the release of compounds is more sustained in the first 60 min and takes 3 h. In addition, it was proven that modified SCLs could be used for three consecutive days with a similar release if SCL was soaked overnight in the drug solution. This fact opens new possibilities of using multitasking solutions as protectors and carriers of drugs, recharging SCLs each night with the selected drug [28]. It is worth noting that using contact lenses for only 30 min a day provides an equivalent IOP control comparable to the regular use of eye drops [149]. Nevertheless, no studies have shown sustained drug release more significant than 3 h, implying the need to invent modified SCLs with better parameters.

It was shown that the addition of Vitamin E to the hydrogel matrix creates a biocompatible diffusion barrier that prolongs drug release over time [150]. In this sense, timolol loaded with Vitamin E-modified lenses showed reasonable control of IOP, requiring only 20% of the dose applied in drops and lasted four days [151].

Similar results were obtained by Hsu et al. [67] using timolol, dorzolamide, or a combination of both. The addition of Vitamin E increased the release of timolol from 1 to 25 h and dorzolamide from 2.5 to 36 h. The equivalent control of intraocular pressure (5 mmHg) was found at concentrations six times lower than those with drops and lasted up to eight days after treatment discontinuation [67].

The same approach was also used for bimatoprost, showing controlled release at therapeutic doses above ten days [66].

However, some limitations of this modification have been shown. While adding an antioxidant to a silicone hydrogel showed an improvement in the drug release parameters [67,151], pHEMA lenses showed no enhancement for timolol or brimonidine after adding Vitamin E [152].

There are also studies indicating the lack of benefits in Vitamin E loading silicone hydrogels [147]. Other studies show that Vitamin E may cause undesirable changes in SCLs physical properties and negatively affect oxygen diffusion and ion permeability [150].

### 9.2. Ocular Allergies

The combination of the ability of contact lenses to act as a physical barrier against the airborne antigen and to serve as a permanent depot of antihistamines may improve the treatment efficacy of some ocular allergic diseases. The authors [98] developed olopatadine affinity SCLs by mimicking the composition of the natural H1 receptor, for which olopatadine acts as a selective antagonist. Functional monomers that correspond to the chemical groups of the receptor and the use of molecular imprinting technology have led to the development of hydrogels capable of loading large amounts of olopatadine and sustaining release upon contact with the tear fluid. Optimised hydrogels prepared with acrylic acid, 2-acrylamide-2-methyl-1-propane-sulfonic acid, and benzyl methacrylate as functional monomers showed a balanced, several-hour release of olopatadine with concentrations similar to those in commercial eye drops. Olopatadine-loaded CL successfully passed the HET-CAM eye irritation test and showed good mast cell compatibility. They succeeded in inhibiting the release of histamine and TNF-α from sensitised mast cells, demonstrating their potential use in preventing and treating allergic conjunctivitis [98].

### 9.3. Antibiotics

Malakooti et al. aimed to develop a drug-soft contact lens suitable for the controlled release of antimicrobial peptides on the ocular surface [95]. Incorporating functional monomers and molecular imprinting techniques to render 2-hydroxyethyl methacrylate (pHEMA) hydrogels able to release polymyxin B and vancomycin in a sustained manner was investigated. Polymyxin B-loaded hydrogels showed good biocompatibility in HET-CAM tests. The functionalised hydrogels also loaded vancomycin and sustained its release, but polymyxin B’s imprint effect was only demonstrated.

Hui et al. studied (in vitro and in vivo) new silicone hydrogel SCLs made using molecular imprinting techniques to extend the release time of the fluoroquinolone antibiotic ciprofloxacin [96].

### 9.4. Antiviral Drugs

Hydrogel contact lenses with an affinity for acyclovir (ACV) and its prodrug valacyclovir (VACV), drugs of the first choice against viral herpes simplex keratitis (HSV), have been designed to ensure the sustained release of therapeutic doses during daily wear. Printed and unprinted hydrogels with different contents of functional methacrylic acid monomer (MAA) were prepared and analysed regarding swelling, transmittance, mechanical properties, and ocular compatibility (hen egg chorioallantoic test (HET-CAM)). Measured values for these parameters were within the range typical of soft contact lenses. Compared to ACV, the charging capacity of the VACV was significantly higher due to the stronger electrostatic interactions with MAAs. The advantages of the printing technology have been proven for VACV [104].

### 9.5. Antifungal Drugs

Fungal keratitis, a disease potentially leading to blindness, is challenging to treat due to the limited number of approved antifungal medications and strict dosing regimens. Thus, the development of SCLs as an antifungal drug delivery platform can improve the treatment of fungal keratitis. SCLs can serve as a drug reservoir to continuously release drugs into the cornea while limiting drug loss through tear drainage, blinking, and nonspecific absorption [14].

### 9.6. Anti-Inflammatory and Immunosuppressive Drugs

White et al. designed molecular imprinted silicone hydrogel contact lenses to simultaneously release up to four matrix molecules, including hydroxypropyl methylcellulose (HPMC), trehalose, ibuprofen, and prednisolone [101].

Naltrexone (NTX) is a potent opioid growth factor receptor (OGFR) antagonist that has proven helpful in treating ocular surface damage. Alvarez-Rivera et al. designed and tested a hydrogel based on 2-hydroxyethyl methacrylate with NTX imprint for sustained drug release on the eye surface. Acrylic acid (AAc) and benzyl methacrylate (BMA) were selected as functional monomers capable of forming binding cavities to mimic the OGFR binding sites for NTX. Printed hydrogels containing functional monomers loaded higher amounts of NTX than unprinted hydrogels by simply soaking in an aqueous drug solution [100].

Cyclosporine A (CyA) was first used in ophthalmology in 1980. It was administered orally after a corneal transplant [100]. Currently, CyA, in the topical form, is a typical treatment for severe dry eye syndrome.

Cheng-Chun Peng et al. [153] tested silicone hydrogel lenses impregnated with CyA to ensure controlled and sustained drug delivery with enhanced bioavailability. Silicone hydrogel lenses impregnated with CyA were compared with CyA loaded 1-DAY ACUVUE contact lenses. Hydrogel lenses have been shown to have a longer drug retention time (up to 15 days) due to CyA’s greater affinity for the gel, high molecular weight, and very high partition coefficient. The larger size and stronger drug binding to the lens polymer reduce the effective diffusivity resulting in a longer release time. The incorporation of vitamin E as a factor prolonging the release time of CyA from hydrogel CLs was also tested. It turned out that the amount of adsorbed CyA increased significantly. Thus, for pure ACUVUE lenses, approximately 60% of the CyA was released within seven days. Lenses without Vitamin E released the drug for 16 and 46 days and retained sufficient oxygen permeability for long-term use. The release time increases to over a month if a 10% Vitamin E supplement is used. It was also shown that the inclusion of Vitamin E enhances the effective partition coefficient of CyA [153].

### 9.7. Diabetes

Silicone-hydrogel SCLs functionalised with bio-inspired chemical groups represent the attempt to design the solution tailored to the needs of diabetic eyes, acting as controlled-release platforms for epalrestat, promoting drug accumulation and diffusion through the cornea. Several sets of silicone hydrogels were synthesised, differing in the content of 2-hydroxyethyl methacrylate (pHEMA), hydroxypropyl terminated mono methacryloxypropyl sym-polydimethylsiloxane (MCS-MC12), and aminopropyl methacrylamide (APMA). Epalrestat was applied before or after polymerisation, and the loading and release profiles were compared. All kits were assessed for optical properties, oxygen permeability, cytological compatibility, ocular surface irritation, and drug penetration through the cornea (using the drug solution as reference). The designed silicone hydrogels showed suitable properties for use as DCRS. Epalrestat-charged hydrogels have also shown anti-cataract activity in an in vitro diabetic eye model [154].

The Table 3, Table 4, Table 5 and Table 6 summarise the results of in vitro experiments examining the use of DDCLs in the context of the desired goals of drug therapy in the treatment of infectious, inflammatory, allergic, and glaucomatous eye diseases [16,155].

## 10. Conclusions

This review shows the development of materials and preparation techniques for therapeutic contact lenses as sustained reservoirs of ophthalmic drugs.

The article provides an overview of available drug modified SCLs preparation methods, such as soaking (immersion coating), molecular imprinting, trapping colloidal nanoparticles containing drugs, drug plate/film, ionic ligand polymer systems, supercritical fluid technology, etc.

The variety of drugs applied to CLs surfaces has also been demonstrated.

Based on the literature review, commercial contact lenses were found to potentially release clinically relevant amounts of anti-infective, anti-inflammatory, antiallergic and anti-glaucoma drugs in vitro. The use of drug-modified commercial CLs improves the bioavailability of medicines in both animals and humans. Lenses loaded with significantly lower drug concentrations than in eye drops have produced similar treatment results. The disadvantage of such a solution is the uncontrolled release of the pharmaceutical.

The drug release kinetics can be improved by including surface-bound molecules such as nanoparticles or liposomes or by the deposition of diffusion barriers such as Vitamin E. The production of tailored drug-eluting materials can also be accomplished by incorporating functional monomers that specifically interact with the drug molecule slowing down its diffusion. The modified material can be prepared by molecular imprinting, changing the ionicity of the material, or by introducing a drug reservoir such as PLGA [16].

As authors of the studies cited here point out, the use of therapeutic SCLs may show the following advantages over conventional eye drops: (1) the contact time of the drug with the precorneal tear film can be prolonged; (2) compliance with frequent and complex dosing regimens can be improved; and (3) less systemic toxicity can be expected due to the optimised amount of drug-loaded to SCLs [6].

There are some limitations to commercialising therapeutic contact lenses. Drug-loading may change critical properties of SCLs and cause low water content, poor tensile strength (mechanical properties), reduced ion permeability, impaired transparency, and decreased oxygen permeability. Other issues that need to be addressed are drug stability during processing/manufacturing (drug integrity test); zero-order release kinetics (burst release prevention); undesirable drug release during the post-production monomer extraction step (to remove unreacted monomers); protein adherence; drug release during storage packaging solution; durability study; cost-benefit analysis [15].

The weak therapeutic points of contact lenses also include possible discomfort and dryness of the eye at the end of the wearing period. This phenomenon has been called “contact lens discomfort” and is considered a common reason for the cessation of contact lens use and a return to conventional drug delivery systems [178]. It is also known that long-term use of contact lenses may be associated with corneal toxicity [179].

Another problem is the unhygienic handling of contact lenses. Thus, improper manipulation may carry a greater risk than instillation of regular eye drops, causing an increased hazard of infection or discomfort during wearing (abandonment of contact lenses), leading to treatment failure [180].

It is also worth noting that administering the drug to contact lenses is not suitable for all patients. Therefore, prior selection should be made.

Another problem is the legal regulations related to the registration of contact lenses as a medical device [48]. It is still not settled whether therapeutic CLs are medical devices, drugs, or a combination of both. There is a large gap between legal issues in this area and the dynamic pace of development of the technologies described.

Currently, the only approved method of preparing therapeutic CLs is the soaking (dip-coating) technique. This method does not meet all the expectations for optimal drug storage. Legal regulations allowing for the expansion of acceptable DCRS are still desirable. Therefore, the process of molecular imprinting comes to the forefront among techniques with the most significant potential in the field of sustainable and adequate ophthalmic drug supply.

The paper shows the pros and cons (Figure 1) of using contact lenses as a convenient ophthalmic medical device. As outlined, it remains a challenge to ensure significantly higher safety, efficacy, and comfort of drug-eluting CLs than conventional ophthalmic formulas.

Patients suffering from eye diseases of various origins can benefit from the emerging DDSCL. However, there is still a need to search for novel medical materials that meet the expectations of both the medical community and patients. It should be emphasised that a required field has been opened [28]. Many DCRS based on SCLs have been developed but regrettably have not been approved by the US FDA until today.

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
