# Peer review of "Soft Contact Lenses as Drug Delivery Systems: A Review"

_molecules, 2021, doi:10.3390/molecules26185577_

Round 1
Reviewer 1 Report
The study is a comprehensive review describing the novelty of soft contact lenses as drug delivery systems. The data is well presented, with elucidative figures 1 and 2, and the reference list is vast.
Some points should be addressed to improve the manuscript:
- Table 2 is unnecessary; it does not bring important information to the manuscript.
- Tables 4 to 7 are very extensive and difficult to follow, particularly the “Results” column. I suggest using keywords or using single sentences to summarize the relevant results from each reference.
- A scheme compiling the advantages, disadvantages, and limitations of the use of SCLs over conventional eye drops would improve the conclusions and the manuscript reading.
- The document needs font size formatting (coherent font size along with the text).
- Revise English language and punctuation
Author Response
Thank you for your valuable comments. The improvements have been made as follows.
Review 1
Some points should be addressed to improve the manuscript:
- Table 2 is unnecessary; it does not bring important information to the manuscript.
Table 2 has been removed.
- Tables 4 to 7 are very extensive and difficult to follow, particularly the “Results” column. I suggest using keywords or using single sentences to summarize the relevant results from each reference.
Tables 1 to 7 have been revised and reworded.
- A scheme compiling the advantages, disadvantages, and limitations of the use of SCLs over conventional eye drops would improve the conclusions and the manuscript reading.
Scheme 1 has been included.
- The document needs font size formatting (coherent font size along with the text).
The font size and type have been checked and formatting when necessary.
- Revise English language and punctuation
English language and punctuation have been revised with Grammarly Premium for MS Office (Grammarly Inc. 2021).
Reviewer 2 Report
In this review paper, the authors described the use of contact lenses as drug delivery systems. In my opinion, the paper can be published after some revisions.
First of all, the template of the journal has been modified. Page headers and footers, such as line numbers, have been canceled from the first pages. In many points of the paper, a larger font than the one contained in the journal template has been used.
In Table 1, a recent review published in an MDPI journal is missing. Please, consider the review “Contact Lenses as Ophthalmic Drug Delivery Systems: A Review” doi: 10.3390/polym13071102.
In subsection 3.2, some references have to be added to give strength to the concepts inserted.
In subsection 3.3, the number of the reference Costa et al. has to be inserted.
The acronym pHEMA has been used in section 4, but it has been defined in section 5. Please, specify all the acronyms when used for the first time.
Section 8.4 is missing.
After the conclusion, the part regarding the authors’ contributions, the declaration of competing interests, and so on is missing.
Author Response
- First of all, the template of the journal has been modified. Page headers and footers, such as line numbers, have been cancelled from the first pages. In many points of the paper, a larger font than the one contained in the journal template has been used.
The font size and typeset have been checked, and the whole text was re-formatted according to the journal template. Also, fonts/sizes used in the figures were unified across the text.
- In Table 1, a recent review published in an MDPI journal is missing. Please, consider the review “Contact Lenses as Ophthalmic Drug Delivery Systems: A Review” doi: 10.3390/polym13071102.
This reference has been added to the submission.
- In subsection 3.2, some references have to be added to give strength to the concepts inserted.
Appropriate, new references have been added to this section.
- In subsection 3.3, the number of the reference Costa et al. has to be inserted.
This reference has been updated according to this remark.
- The acronym pHEMA has been used in section 4, but it has been defined in section 5. Please, specify all the acronyms when used for the first time.
The acronyms have been checked and specified (Table 1 (CLs, DCRS, MAA, DMA, NVP, EGDMA, TRIS, PDMS), (Section 3.3 (beta-CD), Chapter 4 (pHEMA), Section 9.1 (IOP))
- Section 8.4 is missing.
Sections in Chapter 8 have been renumbered.
- After the conclusion, the part regarding the authors’ contributions, the declaration of competing interests, and so on is missing.
Chapter 11 with declaration of competing interests has been added.
Reviewer 3 Report
This is a comprehensive review regarding the recent progress in the application of soft contact lenses as drug delivery systems. The only minor issue I am concerned about is the content of table 1. It's good to provide this, but maybe it's better to include this in the supplementary part as it's not quite related to the major content and it's a big table.
Author Response
- This is a comprehensive review regarding the recent progress in the application of soft contact lenses as drug delivery systems. The only minor issue I am concerned about is the content of table 1. It's good to provide this, but maybe it's better to include this in the supplementary part as it's not quite related to the major content and it's a big table.
Tables 1 to 7 have been revised and reworded.